# Effects of Maternal Chewing on Prenatal Stress-Induced Cognitive Impairments in the Offspring via Multiple Molecular Pathways

**DOI:** 10.3390/ijms21165627

**Published:** 2020-08-06

**Authors:** Qian Zhou, Ayumi Suzuki, Mitsuo Iinuma, Ke-Yong Wang, Kin-ya Kubo, Kagaku Azuma

**Affiliations:** 1Department of Anatomy, School of Medicine, University of Occupational and Environmental Health, 1-1 Iseigaoka, Yahatanishi-ku, Kitakyushu 807-8555, Japan; zhou@med.uoeh-u.ac.jp; 2Department of Pediatric Dentistry, Asahi University School of Dentistry, 1851 Hozumi, Mizuho, Gifu 501-0296, Japan; ayumi@dent.asahi-u.ac.jp (A.S.); iinuma@dent.asahi-u.ac.jp (M.I.); 3Shared-Use Research Center, School of Medicine, University of Occupational and Environmental Health, 1-1 Iseigaoka, Yahatanishi-ku, Kitakyushu 807-8555, Japan; kywang@med.uoeh-u.ac.jp; 4Graduate School of Human Life Science, Nagoya Women’s University, 3-40, Shioji-cho, Mizuho-ku, Nagoya, Aichi 467-8610, Japan; kubo@nagoya-wu.ac.jp

**Keywords:** prenatal stress, chewing, hippocampus, BDNF, glucocorticoid receptor, cognitive impairments

## Abstract

We aimed to investigate the effects of maternal chewing on prenatal stress-induced cognitive impairments in the offspring and to explore the molecular pathways of maternal chewing in a mice model. Maternal chewing ameliorated spatial learning impairments in the offspring in a Morris water maze test. Immunohistochemistry and Western blot findings revealed that maternal chewing alleviated hippocampal neurogenesis impairment and increased the expression of hippocampal brain-derived neurotrophic factor in the offspring. In addition, maternal chewing increased the expression of glucocorticoid receptor (GR) and 11β-hydroxysteroid dehydrogenase isozyme 2 (11β-HSD2) and decreased the expression of 11β-HSD1 in the placenta, thereby attenuating the increase of glucocorticoid in the offspring. Furthermore, maternal chewing increased the expression of 11β-HSD2, FK506-binding protein 51 (FKBP51) and FKBP52 and decreased the expression of 11β-HSD1, thereby increasing hippocampal nuclear GR level. In addition, maternal chewing attenuated the increase in expression of DNMT1 and DNMT3a and the decrease in expression of histone H3 methylation at lysine 4, 9, 27 and histone H3 acetylation at lysine 9 induced by prenatal stress in the offspring. Our findings suggest that maternal chewing could ameliorate prenatal stress-induced cognitive impairments in the offspring at least in part by protecting placenta barrier function, alleviating hippocampal nuclear GR transport impairment and increasing the hippocampal brain-derived neurotrophic factor (BDNF) level.

## 1. Introduction

An increasing body of evidence indicates that the prenatal period is an extremely vital phase for fetal neural development and thus a period of vulnerability for long-lasting and irreversible influences on brain development and behavior. Epidemiologic studies reveal that exposure of pregnant women to a stressful environment is associated with neurodevelopmental disturbances that increase the susceptibility to cognitive impairments, emotional problems, abnormal motor behaviors, language disorders and neuro-immuno-endocrine disturbances in their children [1,2,3]. Findings from animal behavior studies confirm that prenatal stress markedly reduces learning and memory abilities and induces anxiety and depression-like behaviors in the offspring [4,5,6]. Animal morphologic studies demonstrate that prenatal stress also induces structural changes in the offspring hippocampus, including suppressed neurogenesis in the hippocampal dentate gyrus (DG) [7], bidirectional synaptic plasticity impairment, hypomyelination in the hippocampal CA1 region [8,9] and altered hippocampal cell tree size [10], which increases the susceptibility to neurodevelopmental disorders and results in hippocampal-dependent learning and memory impairments [11,12,13]. Thus, ameliorating prenatal stress is an enormous challenge in preventing cognitive impairments in the offspring, and effective methods for coping with prenatal stress are urgently needed.

Chewing as a practical behavior for coping with stress has highly important functions correlated with physical and mental health. In humans, gum chewing alleviates psychologic stress and improves task performance [14,15]. In rodents, chewing during restraint stress alleviates anxiety-like behavior and cognitive impairments induced by restraint stress [16,17]. Maternal chewing also ameliorates prenatal stress-induced spatial learning impairment and deficiencies in hippocampal neurogenesis, myelination and synaptic plasticity in the offspring [18,19,20,21], but the underlying mechanisms have remained unclear.

Glucocorticoid (GC), as the end product of the hypothalamic–pituitary–adrenal (HPA) axis, binds to glucocorticoid receptor (GR), and regulates the stress response [22,23]. Unliganded GR primarily locates in the cytoplasm and translocates to the nucleus upon binding to a ligand [24]. The transport of GR between the cytosol and nucleus mainly depends on FK506-binding protein 51 (FKBP51) and FK506-binding protein 52 (FKBP52), which regulate steroid hormone receptor signaling [25,26]. A markedly abnormal increase in GC disrupts the negative feedback system of the HPA axis, leading to long-lasting and irreversible effects on brain development and behavior and stress-related cognitive disorders [22,27,28,29]. During the fetal period, the placenta forms a barrier to maternal GC depending on the activity of GR and 11β-hydroxysteroid dehydrogenase type 2 (11β-HSD2), which catalyzes the metabolism of GC into inactive metabolites [30,31,32]. The enzyme 11β-hydroxysteroid dehydrogenase type 1 (11β-HSD1), on the other hand, converts inactive GC to its active forms [33,34]. Thus, one aim of the present study was to explore the effect of maternal chewing during prenatal stress on deficiencies in placenta barrier function in dams and hippocampus nuclear GR transport in the offspring induced by prenatal stress.

One important target of GC is brain-derived neurotrophic factor (BDNF), which plays a critical role in regulating hippocampal neurogenesis, synaptic plasticity and learning ability. Both GR and BDNF receptor tropomyosin receptor kinase B are co-expressed in hippocampal neurons [35]. There is a growing body of evidence that GC–BDNF crosstalk is essential for the early-life programming of the HPA axis and neurotrophin signaling [36]. Emerging evidence indicates that epigenetic changes regulate hippocampal memory formation and therefore mediate the development of cognitive disorders induced by prenatal stress [37,38,39], but the mechanisms are not well understood. Hippocampal DNA methylation, histone H3 methylation at lysine 4, 9, 27 and acetylation at lysine 9 are the most well characterized epigenetic alterations affecting learning and memory [40,41,42,43] and might play a crucial role in the behavioral and cognitive disorders in the offspring induced by maternal behavior. As such, the present study also aimed to examine the effects of maternal chewing during prenatal stress on characterized epigenetic alterations in the offspring hippocampus induced by prenatal stress.

In this study, we explored multiple molecular pathways underlying the effects of maternal chewing on ameliorating prenatal stress-induced cognitive disorders in the offspring in a mouse prenatal stress model.

## 2. Results

### 2.1. Maternal Chewing during Prenatal Stress Ameliorated Spatial Learning Impairment in the Offspring Induced by Prenatal Stress

As shown in Figure 1A, the escape latency in the stress group was significantly longer than that in the control group (fourth day: F_2,24_ = 5.142, *p* = 0.014, post hoc, *p* = 0.008; fifth day: F_2,24_ = 5.794, *p* = 0.009, post hoc, *p* = 0.003; sixth day: F_2,24_ = 7.407, *p* = 0.003, post hoc, *p* = 0.001; seventh day: F_2,24_ = 8.751, *p* = 0.001, post hoc, *p* = 0.001), indicating that prenatal stress induced spatial learning impairment in the offspring. The escape latency was significantly shorter in the stress+chewing group compared with the stress group (fourth day: *p* = 0.015; fifth day: *p* = 0.030; sixth day: *p* = 0.013; seventh day: *p* = 0.010), indicating that maternal chewing during prenatal stress ameliorated the prenatal stress-induced spatial learning impairment in the offspring. Performance in the visible probe test did not differ significantly among the three groups, indicating similar motor and visual capabilities among all three groups (Figure 1B, F_2,24_ = 0.023, *p* = 0.977).

### 2.2. Maternal Chewing during Prenatal Stress Ameliorated Hippocampal Neurogenesis Defects in Offspring Induced by Prenatal Stress

Neurogenesis was measured by detecting the expression of doublecortin (a marker for neurogenesis) in the hippocampus. Representative photomicrographs of doublecortin-positive cells in the hippocampus in all three groups are shown in Figure 2A. The number of doublecortin-positive cells in the hippocampal DG in the stress group was markedly lower than that in the control group (F_2,24_ = 8.422, *p* = 0.002, post hoc, *p* = 0.001), indicating that prenatal stress induced hippocampal neurogenesis deficiencies in the offspring. The number of doublecortin-positive cells in the hippocampal DG in the stress+chewing group, however, was significantly greater than that in the stress group (*p* = 0.005), indicating that maternal chewing during prenatal stress alleviated hippocampal neurogenesis deficiency in the offspring (Figure 2B).

### 2.3. Maternal Chewing during Prenatal Stress Increased BDNF Expression in the Hippocampus in the Offspring

Microscopically, immunohistochemical staining revealed lower positive expression of hippocampal BDNF in the stress group compared with the control group and higher positive expression in the stress+chewing group than in the stress group (Figure 3A). Compared with the control group, prenatal stress induced a significant decrease in the mRNA and protein expression of hippocampal BDNF in the offspring (mRNA expression: F_2,24_ = 31.309, *p* < 0.001, post hoc, *p* < 0.001; protein expression: F_2,24_ = 31.309, *p* < 0.001, post hoc, *p* < 0.001; Figure 3B,C). Maternal chewing during prenatal stress significantly increased the mRNA and protein expression of hippocampal BDNF in the offspring (mRNA expression: *p* < 0.001; protein expression: *p* < 0.001).

### 2.4. Maternal Chewing during Prenatal Stress Deceased the GC Level in the Offspring by Protecting the Placenta Barrier Function

As shown in Figure 4A–F, compared with the control group, prenatal stress led to a significant decrease in the mRNA and protein expression of GR and 11β-HSD2 (mRNA expression of GR: F_2,24_ = 40.395, *p* < 0.001, post hoc, *p* < 0.001; protein expression of GR: F_2,24_ = 14.494, *p* < 0.001, post hoc, *p* < 0.001; mRNA expression of 11β-HSD2: F_2,24_ = 25.410, *p* < 0.001, post hoc, *p* < 0.001; protein expression of 11β-HSD2: F_2,24_ = 48.629, *p* < 0.001, post hoc, *p* < 0.001) and a significant increase in the mRNA and protein expression of 11β-HSD1 in the placenta (mRNA expression of 11β-HSD1: F_2,24_ = 43.839, *p* < 0.001, post hoc, *p* < 0.001; protein expression of 11β-HSD1: F_2,24_ = 13.636, *p* < 0.001, post hoc, *p* < 0.001), thereby inducing a significant increase in the serum GC level in the offspring (F_2,24_ = 55.244, *p* < 0.001, post hoc, *p* < 0.001), indicating that prenatal stress induced placenta barrier function impairment. Maternal chewing during prenatal stress protected against the placenta barrier function impairment on the basis of the significant increase in the mRNA and protein expression of GR and 11β-HSD2 and the significant decrease in the mRNA and protein expression of 11β-HSD1 in the placenta (mRNA expression of GR: *p* < 0.001; protein expression of GR: *p* < 0.001; mRNA expression of 11β-HSD2: *p* < 0.001; protein expression of 11β-HSD2: *p* < 0.001; mRNA expression of 11β-HSD1: *p* < 0.001; protein expression of 11β-HSD1: *p* < 0.001), leading to significantly lower serum corticosterone levels in the offspring mice (*p* < 0.001; Figure 4G). These data together indicate that maternal chewing during the prenatal stress period decreased the GC level in the offspring by protecting placenta barrier function (Figure 4H).

### 2.5. Maternal Chewing during Prenatal Stress Ameliorated the Nuclear GR Transport Impairment Induced by Prenatal Stress

Immunohistochemical staining revealed lower positive expression of hippocampal GR in the DG and CA1 in the stress group than in the control group and higher positive expression of hippocampal GR in the DG and CA1 in the stress + chewing group than in the stress group (Figure 5A). As shown in Figure 5B–M, compared with the control group, prenatal stress induced a significant decrease in mRNA, total protein, nucleus protein expression of GR and mRNA and protein expression of 11β-HSD2, FKBP51 and FKBP52 (mRNA expression of GR: F_2,24_ = 38.944, *p* < 0.001, post hoc, *p* < 0.001; total protein expression of GR: F_2,24_ = 33.602, *p* < 0.001, post hoc, *p* < 0.001; nucleus protein expression of GR: F_2,24_ = 36.736, *p* < 0.001, post hoc, *p* < 0.001; mRNA expression of 11β-HSD2: F_2,24_ = 44.405, *p* < 0.001, post hoc, *p* < 0.001; protein expression of 11β-HSD2: F_2,24_ = 15.110, *p* < 0.001, post hoc, *p* < 0.001; mRNA expression of FKBP51: F_2,24_ = 36.641, *p* < 0.001, post hoc, *p* < 0.001; protein expression of FKBP51: F_2,24_ = 48.942, *p* < 0.001, post hoc, *p* < 0.001; mRNA expression of FKBP52: F_2,24_ = 71.790, *p* < 0.001, post hoc, *p* < 0.001; protein expression of FKBP52: F_2,24_ = 20.213, *p* < 0.001, post hoc, *p* < 0.001) and a significant increase in the mRNA and protein expression of 11β-HSD1 in the offspring (mRNA expression of 11β-HSD1: F_2,24_ = 37.964, *p* < 0.001, post hoc, *p* < 0.001; protein expression of 11β-HSD2: F_2,24_ = 89.500, *p* < 0.001, post hoc, *p* < 0.001), indicating that prenatal stress induced a hippocampal nuclear GR transport impairment in the offspring. Maternal chewing during prenatal stress alleviated the hippocampal nuclear GR transport impairment in the offspring as the mRNA and protein expression of 11β-HSD2, FKBP51 and FKBP52 were significantly increased (mRNA expression of GR: *p* < 0.001; total protein expression of GR: *p* < 0.001; nucleus protein expression of GR: *p* < 0.001; mRNA expression of 11β-HSD2: *p* < 0.001; protein expression of 11β-HSD2: *p* = 0.001; mRNA expression of FKBP51: *p* < 0.001; protein expression of FKBP51: *p* < 0.001; mRNA expression of FKBP52: *p* < 0.001; protein expression of FKBP52: *p* < 0.001), and the mRNA and protein expression of 11β-HSD1 were significantly decreased (mRNA expression of 11β-HSD1: *p* < 0.001; protein expression of 11β-HSD2: *p* < 0.001), thereby inducing a significant increase in the hippocampal nucleus protein expression of GR in the offspring.

### 2.6. Maternal Chewing during Prenatal Stress Regulated Enzymes of DNA Methylation as Well as Histone H3 Methylation and Acetylation in the Offspring

As shown in Figure 6, compared with the control group, prenatal stress led to an observable increase in the protein expression of DNMT1 and DNMT3a (protein expression of DNMT1: F_2,24_ = 11.751, *p* < 0.001, post hoc, *p* < 0.001; protein expression of DNMT3a: F_2,24_ = 13.142, *p* < 0.001, post hoc, *p* < 0.001) and an observable decrease in the protein expression of H3K4me3, H3K9me3, H3K27me3 and H3K9ac in the hippocampus of the offspring (protein expression of H3K4me3: F_2,24_ = 5.290, *p* = 0.012, post hoc, *p* = 0.004; protein expression of H3K9me3: F_2,24_ = 12.102, *p* < 0.001, post hoc, *p* < 0.001; protein expression of H3K27me3: F_2,24_ = 7.607, *p* = 0.003, post hoc, *p* = 0.001; protein expression of H3K9ac: F_2,24_ = 6.880, *p* = 0.004, post hoc, *p* = 0.001). Maternal chewing during prenatal stress observably decreased the protein expression of DNMT1 and DNMT3a (protein expression of DNMT1: *p* = 0.008; protein expression of DNMT3a: *p* = 0.001) and observably increased the protein expression of H3K4me3, H3K9me3, H3K27me3 and H3K9ac in the hippocampus of the offspring (protein expression of H3K4me3: *p* = 0.031; protein expression of H3K9me3: *p* = 0.001; protein expression of H3K27me3: *p* = 0.005; protein expression of H3K9ac: *p* = 0.017).

## 3. Discussion

In the present study, we found that prenatal stress increased the blood corticosterone level, decreased the hippocampal GR and BDNF expression and induced hippocampus-dependent cognitive impairments in mouse offspring. Maternal chewing during prenatal stress attenuated elevated circulating corticosterone level, increased the hippocampal GR and BDNF expression and ameliorated cognitive impairments in the offspring induced by prenatal stress.

Consistent with previous reports, allowing pregnant mice to chew on a wooden stick during prenatal restraint stress ameliorated the spatial learning impairment induced in the offspring [18,20,21]. Hippocampal neurogenesis highly modulates cognitive processes such as learning, memory and anxiety by regulating information processing in the hippocampal DG [44,45]. Hippocampal neurogenesis occurs throughout life and is susceptible to internal and external environmental changes, especially during the fetal period [46,47]. Animal studies demonstrate that prenatal restraint stress suppresses hippocampal neurogenesis and results in anxiety- and depressive-like behavior and learning disorders in the offspring [48,49,50]. Our findings that maternal chewing during prenatal stress markedly alleviated the neurogenesis impairment in the hippocampal DG in offspring mice are consistent with previous findings [18,20]. BDNF, as the most abundant neurotrophin in the brain, contributes to enhancing synaptic plasticity and improving cognitive functions such as learning, memory and higher thinking [51]. The higher BDNF and lower GC levels are required for normal neuronal maintenance during the prenatal period. BDNF–GC equilibrium is crucial throughout life as a major mechanism for stress response regulation [36]. Prenatal stress downregulates hippocampal BDNF [18] and GR expression [19]. The combination of low BDNF and low GR expression results in vulnerability to stress-related disorders. The present findings demonstrated that providing pregnant mice with a wooden stick to chew on during prenatal restraint stress attenuated the prenatal stress-induced decrease in hippocampal BDNF and GR expression in the offspring. We consider that maternal chewing could protect against stress-induced steep GC elevation and has beneficial effects that are mediated via upregulating BDNF and GR expression. Prenatal stress induces a conspicuous increase in GC levels in dams, which may expose the fetus to excess GC during a vital period of fetal brain development, producing long-lasting and irreversible effects on neonatal neurodevelopment, neuroendocrine and cognitive function [52,53]. Fetal exposure to excess GC is also associated with increased HPA axis activity and increased serum GC levels in childhood [54,55]. In the present study, prenatal stress led to an increase in GC levels in the mouse offspring, which was attenuated by allowing pregnant mice to chew on a wooden stick during prenatal stress, indicating that maternal chewing during prenatal stress ameliorates the prenatal stress-induced cognitive impairment in the offspring by decreasing GC levels. During development, GC, which is derived from the maternal system via the placenta, is essential for the maturation, development and survival of the fetus. In general, most maternally derived GC is metabolized and only 10–20% of maternal GC passes to the fetus due to the placenta barrier function [56]. The enzyme 11β-HSD2 is thought to provide a barrier function by oxidizing active GC to inactive steroids, protecting the fetus against exposure to excessive GC [57]. Recently, placental GRs were also recognized as key regulators of placenta barrier function to reduce fetal GC levels [58]. Inversely, the enzyme 11β-HSD1 was identified as a reductase, converting inactive GC to its active forms [59]. Here, we demonstrated that prenatal stress markedly decreased the expression of 11β-HSD2 and GR and increased the expression of 11β-HSD1 in the placenta. The abnormal changes were significantly attenuated by maternal chewing during prenatal stress. Accordingly, it seems reasonable to assume that maternal chewing during prenatal stress protects the placenta barrier function against excessive GC induced by prenatal stress in the mouse offspring.

The GR, as a part of the large superfamily of nuclear receptors, is a crucial segment of the HPA axis and regulates hippocampal formation development, structure and functioning by translocating to the nucleus after binding to an agonist [22]. FKBP51 and FKBP52 as chaperone proteins regulate GR transport between the cytosol and nucleus [25,26]. Decreased expression of FKBP51 and FKBP52 leads to nuclear GR transport impairment in the hippocampus, thereby affecting the transcriptional capacity of hippocampal neurons and resulting in HPA axis dysfunction related to anxiety, depressive behaviors and learning and memory impairment [26,60]. Furthermore, the GR distribution between the cytosol and nucleus is also regulated by the intracellular concentration of GC [61]. In the hippocampal formation, the enzyme 11β-HSD1, also known as reductase, converts inactive GC to its active form, whereas the enzyme 11β-HSD2 oxidizes active GC to inactive metabolites [34,62]. The balance between 11β-HSD1 and 11β-HSD2 largely regulates the intracellular GC concentration. In the present study, we found that prenatal stress markedly increased the expression of hippocampal 11β-HSD1 and decreased the expression of hippocampal 11β-HSD2, FKBP51 and FKBP52, thereby decreasing hippocampal nuclear GRs in the offspring, indicating that prenatal stress impairs nuclear GR transport. The impaired nuclear GR transport was ameliorated by maternal chewing during prenatal stress, decreasing the expression of 11β-HSD1 and increasing expression of 11β-HSD2, FKBP51 and FKBP52. Overall, these findings suggest that maternal chewing during prenatal stress ameliorates the impaired nuclear GR transport induced by prenatal stress, at least in part, by upregulating 11β-HSD2, FKBP51 and FKBP52 levels and downregulating 11β-HSD1 levels.

Additionally, the prenatal environment is hypothesized to produce differences in the behavioral phenotypes of the offspring by altering gene expression during the critical period of fetal brain development, and these differences are thought to be associated with changes in histone H3 modification or DNA methylation [63,64]. Evidence indicates that DNA methylation is required for hippocampal function, but excess DNA methylation in the hippocampus is related to spatial memory impairment [65]. The enzymes DNMT1 (a maintenance methyltransferase) and DNMT3a (a de novo methyltransferase) are highly expressed in the brain, where they catalyze DNA methylation, and their levels are likely to reflect the overall level of DNA methylation in the hippocampus. Histone H3 methylation at lysine 4 (H3K4me3) and histone H3 acetylation at lysine 9 (H3K9ac) are considered “active” modifications and are associated with transcriptional activation [66,67], while histone H3 methylation at lysine 9 (H3K9me3) and lysine 27 (H3K27me3) are associated with heterochromatin formation and transcriptional repression, respectively [68]. Increased expression of these genes is necessary for synaptic plasticity in the hippocampus. Our results show that prenatal stress induced an increase in the hippocampal DNMT1 and DNMT3a levels and a decrease in the hippocampal H3K4me3, H3K9me3, H3K27me3 and H3K9ac levels in the offspring. These abnormal changes, however, were significantly attenuated by maternal chewing during prenatal stress. These findings indicate that maternal chewing during prenatal stress is associated with the regulation of the enzymes in hippocampal histone H3 methylation and acetylation as well as DNA methylation. Additionally, a growing body of evidence indicates that maternal adversity stress during pregnancy could lead to epigenetic changes in fetal tissues, which might contribute to heightened HPA reactivity among the offspring [69,70,71]. Further studies are necessary to reveal the underlying mechanisms by which specific genes are selected for the epigenetic regulation of hippocampus-dependent functions.

## 4. Materials and Methods

### 4.1. Animal Models

The experimental protocols used in this work were evaluated and approved by the Ethics Review Committee for Animal Care and Experimentation of the University of Occupational and Environmental Health, Japan (AE 17-013, permission code, 8 June, 2017). Female (*n* = 27) and sexually experienced male (*n* = 27) Institute of Cancer Research (ICR) mice (810– weeks of age) were obtained from Japan SLC (Hamamatsu, Japan) and housed under standard laboratory conditions (temperature: 23 ± 1 °C, humidity: 55% ± 5%, light period: 7:00–19:00, dark period: 19:00–7:00, food and drinking water available ad libitum).

One week after the mice arrived, female mice were placed with male mice for one night, and the next day was specified as gestational day 0. Afterwards, the pregnant mice were singly housed in individual cages and randomly assigned to the CONTROL, STRESS and STRESS+CHEWING groups (*n* = 9/group).

The prenatal restraint stress procedure was performed as previously described [18,20,21]. Briefly, pregnant mice in the STRESS and STRESS + CHEWING groups were placed in a ventilated plastic transparent cylinder (4.5 cm diameter, 10.3 cm long), in which the pregnant mice could move back and forth but not turn around, 3 times daily for 45 min each at 09:00, 13:00 and 17:00, from gestational day 12 until parturition. The pregnant mice in the STRESS+CHEWING group were given a wooden stick (2 mm diameter) placed in front of the nose to chew on during the period of prenatal restraint stress, as previously described [18,20,21]. At the end of each restraint stress procedure, the wooden sticks were examined and counted, and the number of chewed wooden sticks did not differ significantly among the mice in the STRESS + CHEWING group. The pregnant mice in the CONTROL group were neither restrained nor provided with a stick to chew and remained in their home cages.

At weaning (3 weeks after birth), male mice were randomly selected from the CONTROL, STRESS and STRESS + CHEWING groups and assigned to a control, stress or stress + chewing group, respectively (lowercase group names represent offspring groups). Male mice at 1 month of age were used for behavioral, immunohistochemical and molecular experiments, and only one mouse from each litter was used in the same assay. All efforts were made to minimize both the suffering and number of animals used.

### 4.2. Morris Water Maze Test

The Morris water maze test is a behavioral task mostly used with rodents that directly reflects spatial learning ability, and it was performed in this study as previously described [20]. Briefly, a stainless-steel circular pool with a diameter of 90 cm and height of 30 cm was filled with water at a temperature of 23 °C to 23 cm. A mouse was placed in the water at one of four evenly spaced starting locations around the pool and allowed 90 s to locate a platform (12 × 12 cm) hidden just 1 cm underneath the surface. If the mouse failed to find the hidden platform within 90 s, it was manually guided to the platform. Each mouse was given 4 acquisition trials per day for 7 days. A charge-coupled device camera linked to a computer system (Move-tr/2D, Library Co., Ltd., Tokyo, Japan) was used to record the escape latency (the time it took for the mouse to swim from 1 of the 4 starting locations to find and climb onto the platform) and the swim path. On the last day of training, 2 h after the last training trial, the mice were given a probe test in which the platform was visible.

### 4.3. Corticosterone Assay

Blood samples were collected at the canthus of 1-month old male mice between 10:00 and 11:00 a.m. in accordance with Institutional Animal Care and Use Committee (IACUC) guidelines and centrifuged at 3000 rpm for 10 min at 4 °C to separate the sera, and the serum corticosterone levels were detected using an enzyme-linked immunosorbent assay kit, according to the manufacturer’s instructions (Assaypro Co. Ltd., St Charles, MO, USA). Light absorbance was detected by an absorbance microplate reader (Corona Electric Co. Ltd., Ibaraki, Japan).

### 4.4. Real-Time PCR

RNA was extracted from isolated hippocampus or placenta tissues and reverse-transcribed into first-strand cDNA with a GoScript Reverse Transcription System kit (Promega, Madison, WI, USA). The cDNA was used in quantitative polymerase chain reactions (PCR) to assess the mRNA expression of BDNF, GR, 11β-HSD1, 11β-HSD2, FKBP51 and FKBP52 in the hippocampus or placenta tissues. Quantitative real-time PCR analysis was performed on an ABI StepOnePlus System (Applied Biosystems, Foster City, CA, USA) with GoTaq qPCR Master Mix (Promega). The nucleotide sequences of the primers used were as follows: β-actin (forward, *GGAGATTACTGCCCTGGCTCCTA*, reverse, *GACTCATCGTACTCCTGCTTGCTG*); BDNF (forward, *TCATACTTCGGTTGCATGAAGG*, reverse, *ACACCTGGGTAGGCCAAGTT*), GR (forward, *GACTCCAAAGAATCCTTAGCTCC*, reverse, *CTCCACCCCTCAGGGTTTTAT*), 11β-HSD1 (forward, *GGAGCCCATGTGGTATTGACT*, reverse, *CCGCAAATGTCATGTCTTCCAT*), 11β-HSD2 (*GCCCTAGAACTGCGTGACC*, reverse, *AGAACACGGCTGATGTCCTCT*), FKBP51 (forward, *GATGAGGGCACCAGTAACAATG*, reverse, *CAACATCCCTTTGTAGTGGACAT*), FKBP52 (forward, *CCTCTCGAAGGAGTGGACATC*, reverse, *TCCCCGATCATGGGTGTCT*), which were designed using Primer3 software and synthesized at Life Technologies Japan Ltd. (Tokyo, Japan). The mRNA expression levels were normalized with β-actin mRNA expression levels and expressed as relative values (fold-change) to the expression levels in the control mice.

### 4.5. Western Blot Analyses

Protein was extracted from hippocampus or placenta tissue using a cold modified radioimmunoprecipitation assay (RIPA) lysis buffer system (Millipore, Burlington, MA, USA), and nuclear protein and cytoplasmic protein were extracted from the hippocampus using a NucBuster protein extraction kit (Novagen, Darmstadt, Germany). The concentrations of hippocampal protein and placenta protein were detected using a BCA protein assay kit (Thermo Scientific, Waltham, MA, USA). Proteins (30 µg) were separated by sodium dodecyl sulfate-polyacrylamide gel electrophoresis (Invitrogen, Carlsbad, CA, USA) and blotted onto polyvinylidene difluoride membranes (Millipore). After blocking, immunoblotting was performed with the following rabbit polyclonal antibodies at 4 °C overnight: anti-GAPDH (37 kDa, 1:1000), anti-Lamin B1 (68 kDa, 1:1000), anti-BDNF (15 kDa, 1:1000), anti-11β-HSD1 (35 kDa, 1:1000), anti-11β-HSD2(44 kDa, 1:1000), anti-GR (91 kDa, 1:1000), anti-FKBP51 (51 kDa, 1:1000), anti-FKBP52 (56 kDa, 1:1000), anti-DNMT1 (200 kDa, 1:1000), anti-DNMT3a (130 kDa, 1:1000), anti-H3K4me3 (17 kDa, 1:1000), anti-H3K9me3 (17 kDa, 1:1000), anti-H3K27me3 (17 kDa, 1:1000), anti-H3K9ac (17 kDa, 1:1000) and anti-histone H3 (17 kDa, 1:1000). The GAPDH, GR, FKBP51, FKBP52, DNMT1, DNMT3a, H3K4me3, H3K9me3, H3K27me3, H3K9ac and histone H3 antibodies were obtained from Cell Signaling Technology (Danvers, MA, USA), and the Lamin B1, BDNF, 11β-HSD1 and 11β-HSD2 antibodies were obtained from Abcam (Cambridge, UK). The next day, the immunoblotting membranes were incubated in a secondary antibody (1:1000, Cell Signaling Technology, Danvers, MA, USA) for 60 min and then visualized with an ECL kit (GE Healthcare Bio-Science, Chicago, IL, USA). The bands of target protein on the Western blots were measured using an Ez-Capture MG System (Atto Corporation, Tokyo, Japan), and densitometric analyses of the bands were performed using the Scion Image software program (version 4.0.2; Scion Corp., Frederick, MD, USA). In the Western blot analyses, GAPDH was used as a loading control to normalize the protein levels of BDNF, GR, 11β-HSD1, 11β-HSD2, FKBP51, FKBP52, DNA methyltransferase 1 (DNMT1) and DNMT3a, and cytoplasmic protein of GR, Lamin B1, was used as a loading control to normalize the levels of nuclear protein of GR, and histone H3 was used as a loading control to normalize the protein levels of H3K4me3, H3K9me3, H3K27me3 and H3K27ac.

### 4.6. Immunohistochemical Staining

Brain tissues were fixed in 4% neutral buffered paraformaldehyde (pH = 7.4) and embedded in paraffin. The paraffin sections (5 μm) were deparaffinized in xylene and rehydrated in ethanol; then, antigen retrieval was performed by incubating in oiled 0.01 M sodium citrate buffer (pH = 6) and blocking endogenous peroxidase activity using 10% H2O2. To reduce nonspecific staining, the slides were immersed in Protein Block, Serum Free (Dako, Tokyo, Japan) for 15 min. For immunohistochemical staining, the sections (5 μm) were incubated with anti-BDNF (1:100, Abcam, Cambridge, UK), anti-GR (1:400, Cell Signaling Technology, Danvers, MA, USA) or anti-doublecortin (1:1000, Abcam, Cambridge, UK) and then incubated with biotinylated goat anti-rabbit IgG and streptavidin peroxidase complex (Nichirei Biosciences Inc., Tokyo, Japan) for 30 min at room temperature, stained with diaminobenzidine and then counterstained with hematoxylin. A light microscope (Olympus, BX50, Tokyo, Japan) connected to a digital camera was used for examining and photographing the slides. For quantification of doublecortin-positive cells in the DG of both hippocampal lobes, the number of immunopositive cells was counted in 10 randomly selected fields of sections, original magnification ×400, as previously described [72].

### 4.7. Statistical Analysis

All experimental data are expressed as the mean ± SEM. The data were analyzed by one-way analysis of variance (ANOVA), followed by Tukey’s post-hoc test for multiple comparisons between groups, using SPSS software (Version22.0, Chicago, IL, USA). Differences were considered statistically significant at *p* < 0.05.

## 5. Conclusions

The present study demonstrates that maternal chewing during prenatal stress could ameliorate prenatal stress-induced cognitive impairments in the offspring, at least in part, by protecting the placenta barrier function, alleviating hippocampal nuclear GR transport impairment and increasing hippocampal BDNF level.

## Figures and Tables

**Figure 1 ijms-21-05627-f001:**
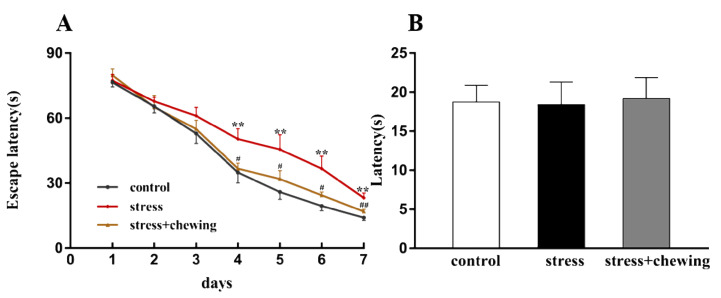
Maternal chewing during prenatal stress ameliorated spatial learning impairments in the offspring. (**A**) The Morris water maze test. (**B**) The visible probe test (** *p* < 0.01 vs. control group, ^#^
*p* < 0.05 vs. stress group, ^##^
*p* < 0.01 vs. stress group, *n* = 9 per group). All data are shown as mean ± SEM.

**Figure 2 ijms-21-05627-f002:**
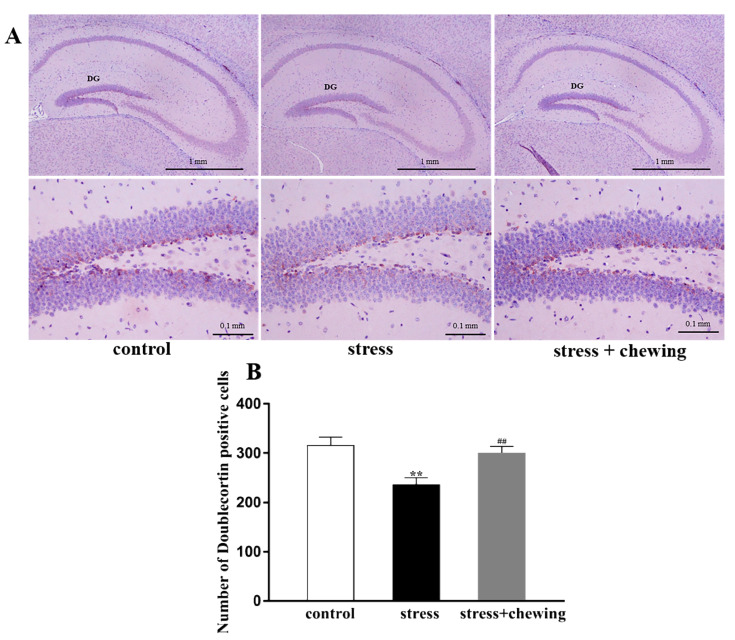
Maternal chewing during prenatal stress ameliorated hippocampal neurogenesis impairments in offspring. (**A**) Photomicrographs showing doublecortin-positive cells in the hippocampus. (**B**) The number of doublecortin-positive cells in the hippocampal dentate gyrus (DG) region (** *p* < 0.01 vs. control group, ^##^
*p* < 0.01 vs. stress group, *n* = 9 per group). All data are shown as mean ± SEM.

**Figure 3 ijms-21-05627-f003:**
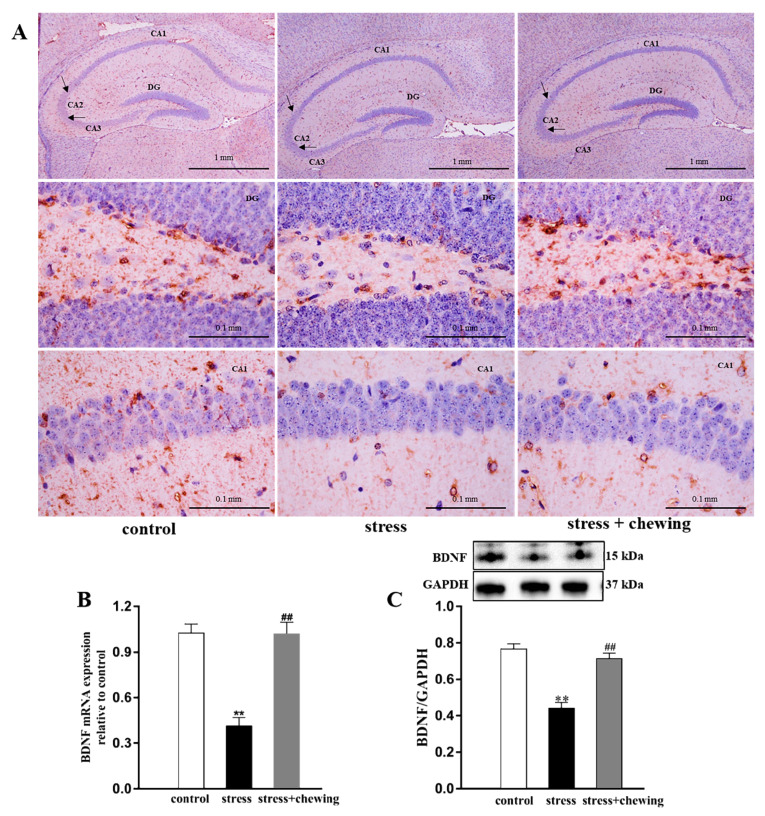
Maternal chewing during prenatal stress increased brain-derived neurotrophic factor (BDNF) expression in the hippocampus in offspring. (**A**) Photomicrographs showing positive BDNF expression in the hippocampus. (**B**,**C**) (** *p* < 0.01 vs. control group, ^##^
*p* < 0.01 vs. stress group, *n* = 9 per group). All data are shown as mean ± SEM.

**Figure 4 ijms-21-05627-f004:**
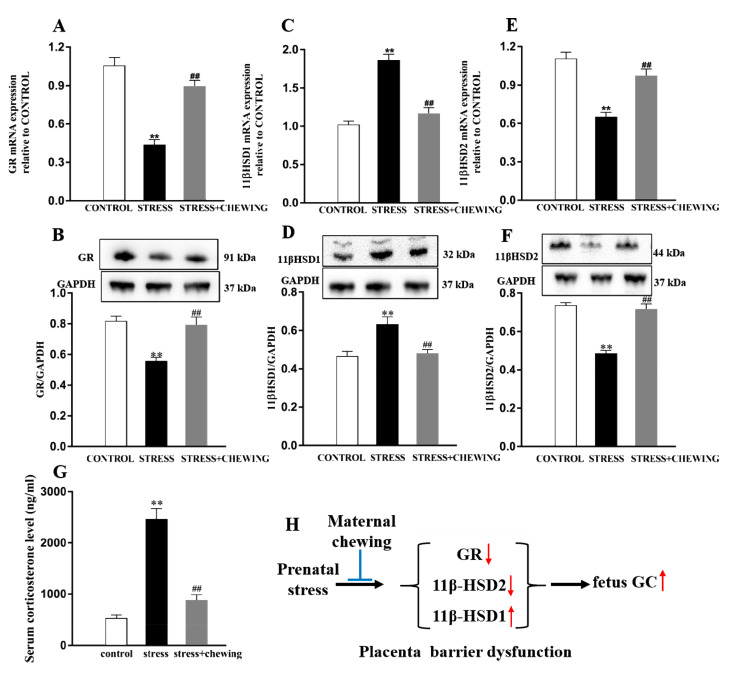
Maternal chewing during prenatal stress deceased the glucocorticoid (GC) level in the mouse offspring by protecting the placenta barrier function. (**A**,**B**) The mRNA and protein expression of glucocorticoid receptor (GR) in the placenta. (**C**,**D**) The mRNA and protein expression of 11β-HSD1 in the placenta. (**E**,**F**) The mRNA and protein expression of 11β-HSD2 in the placenta. (**G**) The serum corticosterone level in the mouse offspring. (**H**) Maternal chewing attenuated prenatal stress-induced increase in GC level in the mouse offspring by protecting placenta barrier function (** *p* < 0.01 vs. control group, ^##^
*p* < 0.01 vs. stress group, *n* = 9 per group). All data are shown as mean ± SEM.

**Figure 5 ijms-21-05627-f005:**
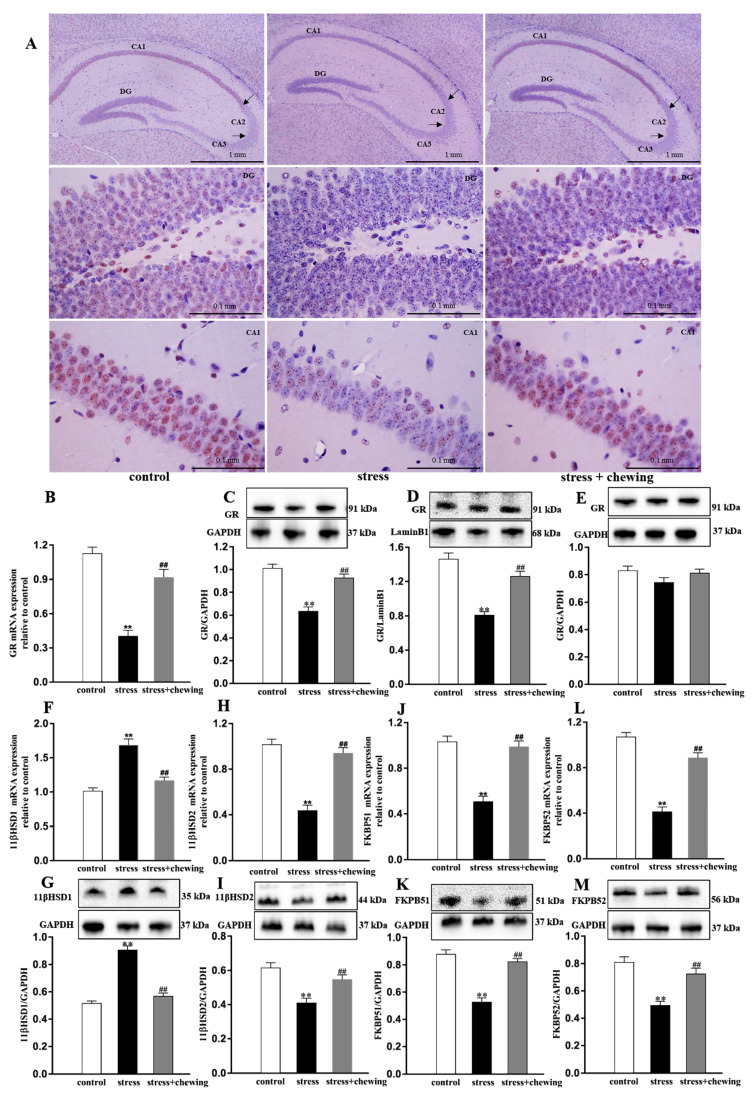
Maternal chewing during prenatal stress ameliorated the nuclear GR transport impairment induced by prenatal stress. (**A**) Photomicrographs showing the positive GR expression in the hippocampus. (**B**–**E**) The mRNA, total protein, nuclear and cytoplasmic protein expression of GR. (**F**,**G**) The mRNA and protein expression of 11β-HSD1. (**H**,**I**) The mRNA and protein expression of 11β-HSD2. (**J**,**K**) The mRNA and protein expression of FKBP51. (**L**,**M**) The mRNA and protein expression of FKBP52 (** *p* < 0.01 vs. control group, ^##^
*p* < 0.01 vs. stress group, *n* = 9 per group). All data are shown as mean ± SEM.

**Figure 6 ijms-21-05627-f006:**
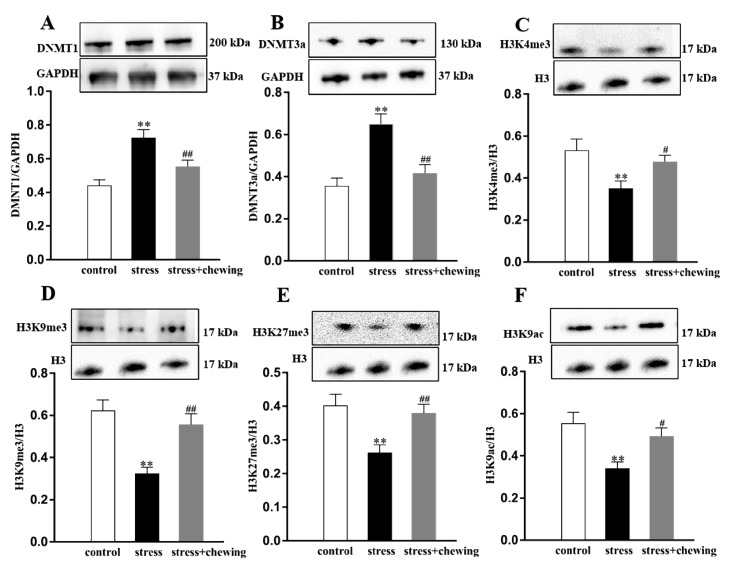
Maternal chewing during prenatal stress regulated enzymes of DNA methylation as well as histone 3 methylation and acetylation in the offspring. (**A**) Protein expression of DNMT1 in the hippocampus in the offspring. (**B**) Protein expression of DNMT3a in the hippocampus in the offspring. (**C**) Protein expression of H3K4me3 in the hippocampus in the offspring. (**D**) Protein expression of H3K9me3 in the hippocampus in the offspring. (**E**) Protein expression of H3K27me3 in the hippocampus in the offspring. (**F**) Protein expression of H3K9ac in the hippocampus in the offspring (** *p* < 0.01 vs. control group, ^##^
*p* < 0.01 vs. stress group, *n* = 9 per group). All data are shown as mean ± SEM.

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
