# Peer review of "Effects of Maternal Chewing on Prenatal Stress-Induced Cognitive Impairments in the Offspring via Multiple Molecular Pathways"

_ijms, 2020, doi:10.3390/ijms21165627_

Round 1
Reviewer 1 Report
Zhou and colleagues examined whether maternal chewing could prevent the effects of prenatal stress on memory in the offspring and the mechanisms by which this occurs. They find that maternal chewing rescues stress-induced memory impairments in spatial memory and changes the expression of various molecular processes in the hippocampus that are known to be involved in memory formation. Overall, the topic is interesting, the methods appropriate and the manuscript is well-written. However, I have a number of concerns that reduce my enthusiasm for the manuscript, particular in terms of a missing control group and the largely correlative nature of the study. Specifically:
Major:
- An important control group is missing as it is unclear if maternal chewing alone has any effect on the molecular processes examined in the hippocampus. As it stands now it is unclear if maternal chewing only affects these processes following a stressor or does so on its own, resulting in a higher baseline that is then reduced some by stress. This is important for interpretation of the results.
- As it currently stands all of the data are correlative and there is no evidence indicating whether maternal chewing alleviates the memory-impairing effects of stress by changing the expression of these various molecular processes in the hippocampus. It is possible that none of mechanisms are involved in the memory enhancing effects of maternal chewing following stress. The statement in the discussion that “The findings of the present study highlight placenta barrier dysfunction, nuclear GR transport impairment, and changes of DNA methylation and histone H3 modifications as critical mechanisms underlying the cognitive impairments induced by prenatal stress in mouse offspring” is significantly overstated and not supported by the data. This is also true for the title and abstract. The manuscript would be significantly strengthened by including direct manipulations of at least one of the many molecular mechanisms examined.
- Related to my above comment, essentially every molecular process examined changed as a function of stress and maternal chewing and it is difficult to understand how all these mechanisms are related. To some extent the molecular aspects of the manuscript read more as an exploratory approach rather than something hypothesis-driven and is largely preliminary in nature.
Additional concerns:
- Were the same mice used for behavioral and molecular experiments? It is not clear from the methods whether these were separate animals. This information needs to be provided.
- The Morris Water Maze data presented in Figure 1 shows clear effects on acquisition of the task. Most studies also include a probe trial in which the platform is removed and the amount of time spent in the target vs adjacent quadrants is quantified, which serves as a long-term memory test. However, the authors did not include one. This would be helpful to show if maternal chewing improves long-term memory following the stress procedure
- The additional of a second hippocampus-dependent task, such as object location or context fear conditioning, would strengthen the behavioral results.
- Note for all figure labeling: The “Chewing” group is actually “Chewing+Stress”, not chewing by itself. Labeling the group as only “Chewing” can be misleading then. I suggest that the authors change this label throughout the manuscript to improve clarification.
- As many different versions of antibodies exist at a given vendor, product numbers should be provided for the antibodies used in the study
- Full statistical analyses should be provided, including degrees of freedom and magnitude of effect
- In Section 2.3 it is not clear why BDNF was examined. Some background information is needed
- Figure 4H: The diagram does not accurately represent the molecular data. As it currently reads, it suggests that maternal chewing alleviates prenatal stress, however, it actually prevents the molecular effects of prenatal stress. The blue line indicating “block” by maternal chewing should instead be on the arrow to the right of prenatal stress.
- For mRNA and western blots, what part of the hippocampus is examined?
- Section 2.6: DNTM1 and DNMT3a expression do not indicate whether DNA methylation levels changed as a result of stress or maternal chewing, rather only provide information about a change in expression of the enzymes involved in this process. Widely used ELISA assays (from Epigentek, etc) would be needed to directly determine if DNA methylation levels changed. Also, DNMT1 and DNMT3a are only involved in one form of DNA methylation (5-mc), though other forms (5-hmc) are also involved in hippocampus-dependent synaptic plasticity.
Author Response
Thank you for your valuable comments. The response to your comments is as follows.
1 An important control group is missing as it is unclear if maternal chewing alone has any effect on the molecular processes examined in the hippocampus. As it stands now it is unclear if maternal chewing only affects these processes following a stressor or does so on its own, resulting in a higher baseline that is then reduced some by stress. This is important for interpretation of the results.
Response: Thank you for your valuable comments. Mastication plays an important role in preserving hippocampus-dependent cognition (Int J Mol Sci 2017). Masticatory dysfunction caused by tooth loss is associated with the hippocampus-dependent cognitive impairment in experimental animals and humans (Int J Mol Sci 2017, Front Physiol 2019).
In humans, gum chewing is reported to relieve stress and improve task performance (Physiol Behav 2009). By using functional magnetic resonance imaging, it was found that chewing increased the blood oxygenation level-dependent signals in various brain regions, including the prefrontal cortex (J Dental Res 2002, Brain Cogn 2013).
In animals, masticatory dysfunction caused by removing teeth results in hippocampus-dependent cognitive impairment (Int J Med Sci 2014, Arch Oral Biol 2017). Chewing has an important role in maintaining some aspects of cognitive function. In this study, we explore the mechanisms underlying the effects of maternal chewing on ameliorating prenatal stress-induced cognitive deficits.
As you indicated, it is important to add a control group for the effects of chewing on non-stressed mice. In fact, we tried to do the experiments with ‘non-stressed chewing’ mice. Unfortunately, it is very difficult to do such experiments with ‘non-stressed chewing’ mice. Mice chew on wooden stick under restraint stress condition. But they cannot chew continuously with free movement around its home cage. So we just compare the findings of the control, stress, and stress+chewing groups.
2 As it currently stands all of the data are correlative and there is no evidence indicating whether maternal chewing alleviates the memory-impairing effects of stress by changing the expression of these various molecular processes in the hippocampus. It is possible that none of mechanisms are involved in the memory enhancing effects of maternal chewing following stress. The statement in the discussion that “The findings of the present study highlight placenta barrier dysfunction, nuclear GR transport impairment, and changes of DNA methylation and histone H3 modifications as critical mechanisms underlying the cognitive impairments induced by prenatal stress in mouse offspring” is significantly overstated and not supported by the data. This is also true for the title and abstract. The manuscript would be significantly strengthened by including direct manipulations of at least one of the many molecular mechanisms examined.
Response: Thank you for your constructive comments. As we can’t do experiments with ‘non-stressed chewing’ mice, we just compared the findings of the intact control, stress, and chewing+stress groups. The results of the present study showed multiple alterations, including placenta barrier, hippocampal GR transport, DNA methylation and H3 modifications. This study indicated that the most direct and important indicator is the change of the blood corticosterone level, which might eventually induce alterations of various molecular pathways, mediated via modification of the glucocorticoid receptor expression. According to your suggestion, we revised some expressions in the title, abstract, discussion and conclusion (Line 2; Line28-33; Line 193-197; Line 397).
3 Related to my above comment, essentially every molecular process examined changed as a function of stress and maternal chewing and it is difficult to understand how all these mechanisms are related. To some extent the molecular aspects of the manuscript read more as an exploratory approach rather than something hypothesis-driven and is largely preliminary in nature.
Response: Thank you for your indication. Numerous previous studies showed that maternal chewing could improve prenatal stress-induced hippocampal neurogenesis, synaptic plasticity and cognitive function by attenuating the stress hormone, corticosterone level (Arch Oral Biol 2019, Int J Med Sci 2018, Brain Res 2016, Neurosci Lett 2014). However, the detailed underlying mechanisms remained unclear. The present study aimed to examine whether maternal chewing during prenatal stress modifies prenatal stress-induced changes in molecular pathways.
The present study indicates that the most direct and important indicator is the change of the blood glucocorticoid (GC) level, which might eventually induce alterations of various molecular pathways, mediated via modification of the glucocorticoid receptor expression.
The present study indicates that the most direct and important indicator is the change of the blood glucocorticoid level, which might eventually induce alterations of various molecular pathways, mediated via modification of the glucocorticoid receptor expression.
During development, GC, which is derived from the maternal system via the placenta, is essential for the maturation, development, and survival of the fetus. In general, most maternally derived GC is metabolized and only 10%-20% of maternal GC passes to the fetus due to placenta barrier function (Cell Mol Life Sci 76, (1), 13-26.,2019). The enzyme 11β-HSD2 is thought to provide a barrier function by oxidizing active GC to inactive steroids, protecting the fetus against exposure to excessive GC (J Clin Endocrinol Metab 100, (4), E542-9.,2015). Recently, placental GR were also recognized as key regulators of placenta barrier function to reduce fetal GC levels (Placenta 54, 24-29.,2017). Inversely, the enzyme 11β-HSD1 is identified as a reductase converting inactive GC to its active forms (Physiol Rev 93, (3), 1139-206.,2013). Here, we demonstrated that prenatal stress markedly decreased the expression of 11β-HSD2 and GR, and increased the expression of 11β-HSD1 in the placenta. The abnormal changes were significantly attenuated by maternal chewing during prenatal stress. Accordingly, it seems reasonable to assume that maternal chewing during prenatal stress protects placenta barrier function against excessive GC induced by prenatal stress in the mouse offspring.
One important target of GC is Brain-derived neurotrophic factor (BDNF), which plays a critical role in regulating hippocampal neurogenesis, synaptic plasticity and learning ability. BDNF-GC equilibrium is crucial throughout life as a major mechanism for stress response regulation. Another important target of GC is glucocorticoid receptor (GR), which is a crucial segment of the HPA axis, and regulates hippocampal formation development, structure, and functioning by translocating to the nucleus after binding to an agonist. The combination of low BDNF and low GR expression favors the vulnerability to develop stress-related disorders. The present findings demonstrated that providing pregnant mice a wooden stick to chew on during prenatal restraint stress attenuated the prenatal stress-induced decrease in the hippocampal BDNF and GR expression in the offspring. We consider that maternal chewing could protect against stress-induced steep GC elevation and has beneficial effects mediated via upregulating BDNF and GR expression.
In conclusion, our results show that maternal chewing during prenatal stress could ameliorate prenatal stress-induced cognitive impairments in the offspring, at least in part by protecting placenta barrier function, alleviating hippocampal nuclear GR transport impairment, and increasing hippocampal BDNF level.
Additional concerns:
4 Were the same mice used for behavioral and molecular experiments? It is not clear from the methods whether these were separate animals. This information needs to be provided.
Response: The behavioral and molecular experiments were performed separately. The Morris Water Maze test was carried out for 7 consecutive days.
5 The Morris Water Maze data presented in Figure 1 shows clear effects on acquisition of the task. Most studies also include a probe trial in which the platform is removed and the amount of time spent in the target vs adjacent quadrants is quantified, which serves as a long-term memory test. However, the authors did not include one. This would be helpful to show if maternal chewing improves long-term memory following the stress procedure
Response: The Morris Water Maze (MWM) is the most commonly used test in study of spatial learning with rodents. The performance measures of animals in MWM include the escape latency, visible probe, a probe trial retention test, dwell time in each of the pool's quadrants, and the total number of times the rodents crossed the original platform location. In this study, we carried out the classic MWM test including the escape latency and visible probe. We consider to perform the other trials in the near future to evaluate the long-term memory in mice comprehensively.
6 The additional of a second hippocampus-dependent task, such as object location or context fear conditioning, would strengthen the behavioral results.
Response: Thank you for your suggestion. There are many neurobehavioral tasks for examining hippocampus-dependent functions. We would like to assess the mouse cognition in active place avoidance task, object location test, and contextual fear conditioning in our next study.
7 Note for all figure labeling: The “Chewing” group is actually “Chewing+Stress”, not chewing by itself. Labeling the group as only “Chewing” can be misleading then. I suggest that the authors change this label throughout the manuscript to improve clarification.
Response: Thank you for your suggestion. The word ‘chewing’ was changed to ‘stress+ chewing’ in our manuscript, including all figures.
8 As many different versions of antibodies exist at a given vendor, product numbers should be provided for the antibodies used in the study.
Response: We provided the product numbers of all antibodies used for immunohistochemistry and western blot in this study by attachment.
9 Full statistical analyses should be provided, including degrees of freedom and magnitude of effect.
Response: As you indicated, we provide the full statistical data, including degrees of freedom and magnitude of effect by attachment.
10 In Section 2.3 it is not clear why BDNF was examined. Some background information is needed.
Response: Thank you for your indication. One important target of glucocorticoid (GC) is brain-derived neurotrophic factor (BDNF), which plays a critical role in regulating hippocampal neurogenesis, synaptic plasticity and learning ability. Both glucocorticoid receptor (GR) and BDNF receptor, Tropomyosin receptor kinase B (TrkB) are co-expressed in hippocampal neurons (Neuroscience 239:173-195, 2013). There is a growing body of evidence that GC-BDNF crosstalk is essential for the early-life programming of the HPA axis and neurotrophin signaling (Front Mol Neurosci 2015). We added the following three sentences of the BDNF background information in the Introduction section, as you suggested.
One important target of GC is brain-derived neurotrophic factor (BDNF), which plays a critical role in regulating hippocampal neurogenesis, synaptic plasticity and learning ability. Both GR and BDNF receptor, tropomyosin receptor kinase B are co-expressed in hippocampal neurons (Neuroscience 239:173-195, 2013). There is a growing body of evidence that GC-BDNF crosstalk is essential for the early-life programming of the HPA axis and neurotrophin signaling (Front Mol Neurosci 2015).
In the Discussion section, we also added the following paragraph.
The higher BDNF and lower GC levels are required for normal neuronal maintenance during the prenatal period. DNF-GC equilibrium is crucial throughout life as a major mechanism for stress response regulation (Front Mol Neurosci 2015). The prenatal stress downregulates the hippocampal BDNF (Kubo et al., 2018) and GR expression (Kubo et al., 2019). The combination of low BDNF and low GR expression favors the vulnerability to develop stress-related disorders. The present findings demonstrated that providing pregnant mice a wooden stick to chew on during prenatal restraint stress attenuated the prenatal stress-induced decrease in the hippocampal BDNF and GR expression in the offspring. We consider that maternal chewing could protect against stress-induced steep GC elevation and has beneficial effects mediated via upregulating BDNF and GR expression.
11 Figure 4H: The diagram does not accurately represent the molecular data. As it currently reads, it suggests that maternal chewing alleviates prenatal stress, however, it actually prevents the molecular effects of prenatal stress. The blue line indicating “block” by maternal chewing should instead be on the arrow to the right of prenatal stress.
Response: Thank you for your indication. We revised the figure 4H according to your suggestion.
- For mRNA and western blots, what part of the hippocampus is examined?
Response: The mouse hippocampus proper includes the dentate gyrus and cornu ammonis (CA) fields (CA1-CA3). In this study, we examined the mRNA and protein expressions of the whole hippocampus proper includes the dentate gyrus and CA fields.
13 Section 2.6: DNTM1 and DNMT3a expression do not indicate whether DNA methylation levels changed as a result of stress or maternal chewing, rather only provide information about a change in expression of the enzymes involved in this process. Widely used ELISA assays (from Epigentek, etc) would be needed to directly determine if DNA methylation levels changed. Also, DNMT1 and DNMT3a are only involved in one form of DNA methylation (5-mc), though other forms (5-hmc) are also involved in hippocampus-dependent synaptic plasticity.
Response: Thank you for your indication. In this study, we examined the DNTM1 and DNMT3a expression by Western blot. Our results showed that prenatal stress induced an increase in the hippocampal DNMT1 and DNMT3a levels, and these abnormal changes were significantly attenuated by maternal chewing during prenatal stress. These findings indicate that maternal chewing during prenatal stress are associated with the regulation of hippocampal DNA methylation. It is better to measure DNA methylation levels using ELISA method. Thank you again for your valuable suggestion and telling us the information about ELISA assays, we will pay more attention to DNA methylation and determine DNA methylation level using ELISA assay in subsequent experiments.
Reviewer 2 Report
The manuscript "Maternal Chewing during Prenatal Stress Ameliorates Prenatal Stress-induced Cognitive Impairments in the Offspring via Multiple Molecular" is a very interesting piece of information that begins to clarify how the maternal stress can influence the genome in the offspring.
It focusses on the aspect that physical exercise is a trigger for the production of neurotrophins and for the modulation of neuro-inflammatory reactions, playing also a role in preserving cognitive function in aging and neuropathological conditions. It further agrees with previous multidisciplinary studies suggesting that masticatory dysfunction may cause hippocampal function impairment leading to deficits in spatial memory and learning. Overall the paper is well written and scientifically sound, photographs are superb, however there are some issues that the authors may draw attention to.
Introduction:
as there is ample evidence that by exercising muscles it can be affected the expression of brain-derived neurotrophic factor synthesis in the dentate gyrus of the hippocampus, where the trophin appears also implicated in regulating the neurogenesis, perhaps the authors should include a short text to introduce the BDNF itself as to justify its choice among the markers chosen for the present study.
Results and figures:
please refer to each analysis and each x-axis group in the graphs to 'stress + chewing'.
Discussion
in light of the perspective of obtaining further insights onto the very linkage between maternal chewing and amelioration of prenatal stress-induced cognitive impairments in the offspring, authors should discuss better the role of the sole chewing in the cognitive improvement observed in the offspring: do the markers change at the same way as the stress + chewing?
Author Response
Thank you for your valuable comments. The response to your comments is as follows.
- Introduction: as there is ample evidence that by exercising muscles it can be affected the expression of brain-derived neurotrophic factor synthesis in the dentate gyrus of the hippocampus, where the trophin appears also implicated in regulating the neurogenesis, perhaps the authors should include a short text to introduce the BDNF itself as to justify its choice among the markers chosen for the present study
Response: Thank you for your suggestion. We added one paragraph of the BDNF background information in the Introduction and Discussion sections (Line 75-79; Line 209-218).
- Results and figures: please refer to each analysis and each x-axis group in the graphs to 'stress + chewing'.
Response: According to your suggestion, the word ‘chewing’ was changed to ‘stress+ chewing’ in our manuscript, including all figures.
- Discussion: in light of the perspective of obtaining further insights onto the very linkage between maternal chewing and amelioration of prenatal stress-induced cognitive impairments in the offspring, authors should discuss better the role of the sole chewing in the cognitive improvement observed in the offspring: do the markers change at the same way as the stress+chewing?
Response: it is important to add a control group for the effect of chewing on non-stressed mice. In fact, we tried to do the experiments with ‘non-stressed chewing’ mice. Unfortunately, it is very difficult to do such experiments with ‘non-stressed chewing’ mice. Mice chew on wooden stick under restraint stress condition. But they cannot chew with free movement around its home cage. So we just compare the findings of the control, stress, and stress+chewing groups.
Round 2
Reviewer 1 Report
The authors have done a nice job addressing my previous concerns. The exclusion of the control experiment is understandable considering the authors’ explanation and the claims made have been toned down to better reflect the correlative nature of the study. I have just a few minor comments:
- The authors have gone above and beyond in providing statistical analyses and I applaud the transparency. However, the supplemental file including this information is never referenced anywhere in the text. A reference should be included so the reader knows to look there.
- If the authors don’t want to include confirmation of actual DNA methylation, then they need to tone down their conclusions regarding this. DNA methylation is a very stable process and changes in the expression of DNMT1 and DNMT3a could or could not be related to changes in DNA methylation. Due to this, the authors can say there were effects on DNA methylation enzymes, but not on DNA methylation per se. (see discussion line 276, Section 2.6 and Figure 6 legend)
- Figure 5 is distorted in the new PDF. All of the graphs are darkened out and cannot be read. This could be an issue with the website, not the author’s file, but should be corrected.
Author Response
Responses to the comments of Reviewer 1
Thank you for your valuable comments. The response to your comments is as follows.
- The authors have gone above and beyond in providing statistical analyses and I applaud the transparency. However, the supplemental file including this information is never referenced anywhere in the text. A reference should be included so the reader knows to look there.
Response: Thank you for your kind comments. We added the main statistical data including degrees of freedom and p values in the Results section, according to your suggestion.
- If the authors don’t want to include confirmation of actual DNA methylation, then they need to tone down their conclusions regarding this. DNA methylation is a very stable process and changes in the expression of DNMT1 and DNMT3a could or could not be related to changes in DNA methylation. Due to this, the authors can say there were effects on DNA methylation enzymes, but not on DNA methylation per se. (see discussion line 276, Section 2.6 and Figure 6 legend)
Response: Thank you for your indication. The expression of ‘DNA methylation’ was changed to ‘enzymes of DNA methylation’ in the section 2.6, Figure 6 legend, and Discussion line 311.
- Figure 5 is distorted in the new PDF. All of the graphs are darkened out and cannot be read. This could be an issue with the website, not the author’s file, but should be corrected.
Response: Figure was corrected, as you indicated.